# Towards LLM-driven Dialogue State Tracking

**Yujie Feng    Zexin Lu    Bo Liu    Liming Zhan    Xiao-Ming Wu***

Department of Computing, The Hong Kong Polytechnic University, Hong Kong S.A.R.

{yujie.feng, zexin.lu, bokelvin.liu, lmzhan.zhan}@connect.polyu.hk

xiao-ming.wu@polyu.edu.hk

## Abstract

Dialogue State Tracking (DST) is of paramount importance in ensuring accurate tracking of user goals and system actions within task-oriented dialogue systems. The emergence of large language models (LLMs) such as GPT3 and ChatGPT has sparked considerable interest in assessing their efficacy across diverse applications. In this study, we conduct an initial examination of ChatGPT's capabilities in DST. Our evaluation uncovers the exceptional performance of ChatGPT in this task, offering valuable insights to researchers regarding its capabilities and providing useful directions for designing and enhancing dialogue systems. Despite its impressive performance, ChatGPT has significant limitations including its closed-source nature, request restrictions, raising data privacy concerns, and lacking local deployment capabilities. To address these concerns, we present LDST, an LLM-driven DST framework based on smaller, open-source foundation models. By utilizing a novel domain-slot instruction tuning method, LDST achieves performance on par with ChatGPT. Comprehensive evaluations across three distinct experimental settings, we find that LDST exhibits remarkable performance improvements in both zero-shot and few-shot setting compared to previous SOTA methods. The source code[1] is provided for reproducibility.

## 1  Introduction

Task-oriented dialogue systems have emerged as powerful tools for assisting users in accomplishing a wide range of tasks (Huang et al., 2020). These systems, such as Apple Siri and Microsoft Cortana, function as virtual personal assistants, providing support for tasks like flight reservations, appointment scheduling, and hotel bookings. Dialogue State Tracking (DST) plays a crucial role in task-oriented dialogue systems by accurately tracking

---
*Corresponding author.

[1] https://github.com/WoodScene/LDST

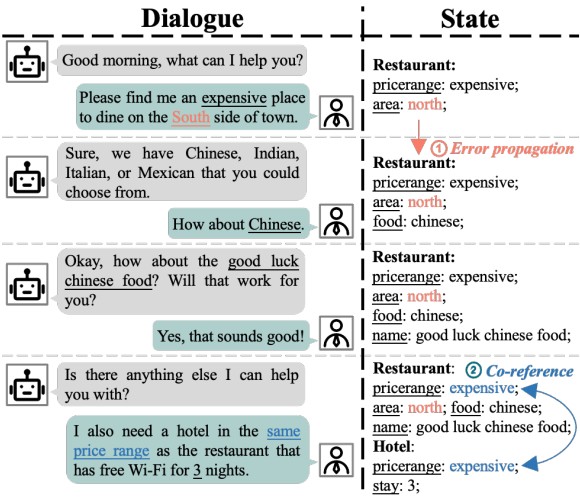

Figure 1: Example of a multi-domain dialogue. The slots "hotel-pricerange" and "restaurant-pricerange" have a co-reference relationship, where the value of the former is inferred from the latter. The slot "restaurant-area" demonstrates error propagation behavior.

the evolving user goals and system actions during a conversation. In general, the multi-domain dialogue state is represented as a list of triplets in the form of (domain, slot, value), e.g., "<restaurant, area, east>". These predefined slot pairs are extracted from the dialogue context at each turn.

A plethora of models have been proposed to address the challenges of multi-domain DST, as documented in recent studies (Qixiang et al., 2022; Zhou et al., 2022; Feng et al., 2022b; Guo et al., 2022a; Yang et al., 2022; Ma et al., 2023; Xu et al., 2023a). These models primarily focus on effective transfer and generalization across diverse domains, addressing the crucial challenges of co-reference (Feng et al., 2022a) and error propagation problem (Wang and Xin, 2022) depicted in Figure 1. The co-reference challenge poses a significant hurdle in enhancing DST performance, as it arises from the linguistic variations in multi-turn dialogues where slots and values are often indirectly

expressed. Moreover, the error propagation issue emerges when the model fails to recognize and rectify errors in the previously predicted dialogue state, leading to the persistence of errors in subsequent turns. Despite significant efforts to address these issues, they persist as ongoing challenges.

In recent days, the emergence of large-scale pre-trained language models has revolutionized the field of natural language processing (NLP). Models like ChatGPT[2] have shown excellent performance, sparking significant interest in evaluating their effectiveness across different dimensions (Tan et al., 2023; Wang et al., 2023; Jiao et al., 2023; Yang et al., 2023a; Gao et al., 2023; Liu et al., 2023). Despite the significant advancements made by large language models (LLMs), their performance in multi-domain DST remains relatively unexplored. To bridge this research gap, we conduct an evaluation of ChatGPT's capabilities for DST. The evaluation unveils ChatGPT's exceptional performance in the DST task, offering valuable insights to researchers and providing useful directions for further exploration.

While ChatGPT demonstrates superb performance, it has significant limitations (Zhou et al., 2023; Yang et al., 2023a; Cao et al., 2023). Firstly, it is not open source, so the underlying code and model parameters cannot be modified by users. Second, it is subject to request limitations, which can restrict its usage in high-demand scenarios. Furthermore, there are concerns regarding strong data privacy protection, as the system may collect and store user data. Lastly, ChatGPT cannot be deployed locally, limiting its availability and control. These limitations hinder the applicability and adoption of ChatGPT in various practical scenarios for building task-oriented dialogue systems.

To overcome the limitations of ChatGPT, we introduce LDST, a DST framework driven by LLMs but based on smaller, open-source foundation models. LDST employs a novel assembled domain-slot instruction tuning method and a parameter efficient tuning technique, enabling it to achieve performance comparable to ChatGPT while utilizing a much smaller model and limited computational resources. LDST demonstrates exceptional performance across three different experimental settings, surpassing prior state-of-the-art methods by a large margin and demonstrating its remarkable adaptability and generalization capabilities. Our main

[2]https://chat.openai.com

contributions are concluded as follows:

- We present the first evaluation of ChatGPT in DST task, highlighting its superior performance over prior methods and providing valuable insights for advancing dialogue systems.

- We propose LLM-driven DST (LDST) based on smaller, open-source foundation models. LDST achieves comparable performance to ChatGPT by employing an innovative assembled domain-slot instruction tuning technique.

- We extensively evaluate LDST on three benchmark datasets across various experimental settings, revealing significant performance improvements over previous approaches. In the zero-shot scenario, LDST boosts the JGA score by 16.9%, elevating it from 65.3% to an outstanding 82.2%. In the few-shot scenario, LDST improves the JGA score by 7.5%, raising it from 47.7% to a notable 55.2%.

## 2 Assessing the Capabilities of ChatGPT for DST

In this section, we evaluate the effectiveness of ChatGPT in addressing the DST task. Before going into detail, we first formally define the problem.

**DST: Problem Formulation**   In task-oriented dialogue systems, a dialogue with $T$ turns of conversations between the system and the user can be represented as $\{(A_1, U_1), (A_2, U_2) \ldots, (A_T, U_T)\}$, where $A$ represents system response and $U$ represents user input. A predefined slot set $\mathcal{S} = \{S_1, \ldots, S_J\}$ is given, where $J$ is the total number of slots. The dialogue context at turn $t$ includes previous turns of interactions, denoted as $\mathcal{X}_t = \{(A_1, U_1), (A_2, U_2) \ldots, (A_t, U_t)\}$. The dialogue state at turn $t$ is represented as a set of (slot, value) pairs, denoted as $\mathcal{B}_t = \{(S_1, V_1^t), \ldots, (S_J, V_J^t)\}$, where $V_J^t$ is the value of slot $S_J$. For multi-domain DST, following previous works (Lee et al., 2019), a slot is defined as the concatenation of the specific domain and the slot, e.g., "<restaurant-area>". If no information is provided in the dialogue about a specific slot, the value associated with that slot is set to "NONE". Essentially, the DST problem is defined as learning a dialogue state tracker $\mathcal{F} : \mathcal{X}_t \to \mathcal{B}_t$.

**Leveraging ChatGPT for DST**   We evaluate the performance of ChatGPT (using the gpt-3.5-turbo API service) on three multi-domain DST benchmarks, using the JGA and AGA evaluation metrics

| | Prompt Type | | Input | Expected Output | AGA |
|---|---|---|---|---|---|
| 1 | Single return | No demo | Perform the task of multi-domain dialogue state tracking.
The following is the dialogue you need to test: [dialogue context]
Please return the value of slot: <hotel-pricerange>. [the description of the slot and its possible value list]
So the value of slot <hotel-pricerange> is | hotel-pricerange: Cheap | 95.92 |
| 2 | Multi return | No demo | Perform the task of multi-domain dialogue state tracking.
And the slot schema is in this list :
['hotel-pricerange', 'train-bookpeople', 'train-leaveat', 'train-destination', 'restaurant-area', xxx].
The following is the dialogue you need to test: [dialogue context]
Please return the value of slot list [hotel-pricerange, train-bookpeople, train-leaveat, ...]. | hotel-pricerange: Cheap,
train-bookpeople: 2,
train-leaveat: 14:00,
… | 81.50 |
| 3 | Single return | One demo | Perform the task of multi-domain dialogue state tracking.
I will show you an example. Please return to the state of the slot.
The example dialog is: [dialogue context]
So the value of slot <hotel-pricerange> is
Output result: cheap
The following is the dialogue you need to test: [dialogue context]
Please return the value of slot: <hotel-pricerange>. [the description of the slot and it's possible value list]
So the value of slot <hotel-pricerange> is | hotel-pricerange: Cheap | 91.93 |
| 4 | Multi return | One demo | Perform the task of multi-domain dialogue state tracking.
I will show you an example. Please return to the state of the slot list: ['hotel-pricerange', 'train-bookpeople', xxx]
The example dialog is: [dialogue context]
Output result: Train-Departure:Norwich, Train-Arrival: Cambridge, hotel-pricerange: Cheap, xxx
The following is the dialogue you need to test: [dialogue context]
Please return the value of slot list [hotel-pricerange, train-bookpeople, train-leaveat, ...]. | hotel-pricerange: Cheap,
train-bookpeople: 2,
train-leaveat: 14:00,
… | 73.33 |

Figure 2: Illustration of different prompt templates used and the corresponding results on the MultiWOZ 2.2 test set.

(for detailed descriptions of the datasets and metrics, refer to Section 4.1). As shown in Figure 2, we explore various prompt templates and utilize the MultiWOZ 2.2 dataset (Zang et al., 2020) to select the optimal prompt.

In Figure 2, "single return" and "multi return" refer to the number of slot values returned in each ChatGPT API request. "Single return" involves requesting and receiving values for one slot at a time, while "multi return" entails requesting and receiving values for all slots simultaneously. For instance, in the MultiWOZ 2.2 dataset which has 49 different slots, "multi return" retrieves values for all 49 slots in a single request. This causes a significant increase API requests for "single return" but simplifies the model's task, resulting in improved performance. Conversely, "multi return" reduces API requests but increases token count per request. "No/one demo" denotes whether an example is provided in the prompt as a demonstration to aid the model's comprehension of the task. Selecting "one demo" is similar to adopting the in-context learning concept. Detailed prompt template design is provided in the Appendix A.1.

**Performance of ChatGPT** As can be seen from Figure 2, the first prompt, which retrieves the value of a single slot in each request without including a demo in the input, achieves the highest AGA score. This is attributed to the inherent difficulty of the task that necessitates the model to provide multiple slot values in a single request. We have observed that ChatGPT tends to predict "NONE" for slots that should have a specific value. For instance, in the case of the slot "hotel-leaveat" where

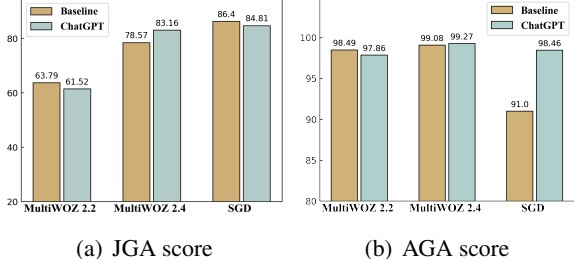

(a) JGA score      (b) AGA score

Figure 3: The results of the best baseline and ChatGPT on various datasets. The higher the values of the JGA and AGA metrics, the better. SOTA results for Multiwoz 2.2, Multiwoz 2.4, JGA score for SGD datasets, and AGA score for SGD datasets were obtained from previous works (Bang et al., 2023a; Ye et al., 2022a; Zhao et al., 2022; Feng et al., 2022a), respectively.

the expected value is "14:00", ChatGPT may incorrectly predict "NONE", resulting in lower prediction accuracy. Secondly, the addition of a demo to the input has a reduced effect, which may seem counter-intuitive. However, our analysis of the error results suggests that ChatGPT also analyzes the dialogue context within the demo, even when the demo and tested sample are clearly differentiated in the input. Therefore, we chose the first prompt as the best template for the subsequent evaluation.

The full evaluation results of ChatGPT on the three datasets[3] are shown in Figure 3. Firstly, on the SGD dataset, the AGA score of ChatGPT is significantly superior than the previous SOTA method (Feng et al., 2022a), and it achieves a 7.46% ab-

---

[3] The evaluation of the MultiWOZ 2.2 dataset were conducted between April 15th and 18th, 2023. The evaluations of MultiWOZ 2.4 occurred between June 10th and 12th, 2023. The SGD was assessed between June 14th and 17th, 2023.

solute imporvement in AGA score. In addition, the average improvement on the three datasets is 0.73% in JGA score and 2.34% in AGA score. Secondly, ChatGPT's performance on the Multi-WOZ 2.2 dataset is slightly worse than the previous SOTA method (Bang et al., 2023a). However, through careful analysis of the errors, we found that 70% of them were due to annotation errors in the original dataset. Thus, on the MultiWOZ 2.4 dataset which has fixed the annotation errors, ChatGPT outperforms the best baseline method (Ye et al., 2022a).

**Limitations of ChatGPT** In summary, ChatGPT exhibits comparable performance when solving the DST task compared to the previous SOTA methods. This highlights the ability of current LLMs to capture and comprehend complex linguistic patterns and dependencies within multi-turn dialogues. However, ChatGPT has several significant limitations that impede its applicability and adoption in various practical scenarios. Firstly, we observed that ChatGPT often provides responses with a significant amount of explanatory content, or it may not align perfectly with our expected answer format. For instance, when the ground truth value is "2 pm," ChatGPT might return "14:00." While both are essentially correct answers, such variations can affect the accuracy of the final metrics. And Chat-GPT is not open source, which restricts the ability of developers and researchers to modify and customize the model. Secondly, ChatGPT is subject to request limitations, which may impact real-time or high-volume applications that rely on prompt and efficient responses. Furthermore, ChatGPT operates in a cloud-based environment and lacks strong data privacy protection measures, which raises concerns about the privacy and security of sensitive information shared during the dialogue sessions. Lastly, ChatGPT relies on an internet connection for operation and cannot be deployed locally.

## 3 Fine-tuning Smaller Foundation Models with Instructions for DST

To overcome the aforementioned limitations of ChatGPT, we introduce LDST, an LLM-driven DST framework that leverages fine-tuning smaller, open-source foundation models such as LLaMa (Touvron et al., 2023) with instructions specially tailored for DST. We first outline the process of constructing an instruction dataset for the multi-domain DST task. Next, we utilize a

parameter-efficient fine-tuning (PEFT) technique to train the foundation model with the instruction dataset. PEFT enables the training of a foundation model with limited computational resources.

### 3.1 Instruction Tuning

Unlike prompt tuning, instruction tuning (Chung et al., 2022) provides more explicit and detailed guidance to the model through task-specific instructions. This allows for finer control over the model's behavior and leads to improved performance compared to prompt tuning. The core idea of instruction tuning is designing the instruction dataset, typically including instruction, input, and output fields. Usually, different instructions are assigned for different tasks. However, employing a fixed instruction template for multi-domain DST may limit the model's robustness, as emphasized by Wang et al. (2023), which highlights the crucial influence of prompt design on model performance.

To address this challenge, we propose a novel Assembled Domain-Slot Instruction Generation approach for the DST task. This approach generates diverse instruction samples by randomly combining different instruction and input templates, exposing the model to a rich variety of instruction types during the fine-tuning process to reduce the model's sensitivity to prompts. As shown by the provided example in Figure 4, for each sample in the original dataset, it consists of the dialogue context $\mathcal{X}_t$ at turn $t$, the current requested slot $S_J$ and its corresponding state $V_J^t$. The raw data is then passed through the Instruction Data Generation module to generate instruction samples. The detailed template settings for each field are introduced as follows.

**Instruction Prompt** Specifically, two types of instruction templates are defined: (1) Standard Slot Tracking Instruction and (2) Customized Slot Tracking Instruction. The difference between them is that the Customized Slot Tracking Instruction provides a more specific domain-slot information. And the instruction field of each sample is randomly selected from these two templates.

**Input Prompt** For the input field, the prompt template is composed of four main parts: (1) the dialogue context, (2) domain-slot description prompt, (3) Possible Value List (PVL) prompt and (4) the query prompt. The green, purple, blue and orange text in the example in Figure 4 refers to these four prompts respectively. In particular, we concatenate all sub-sequences with special segment tokens,

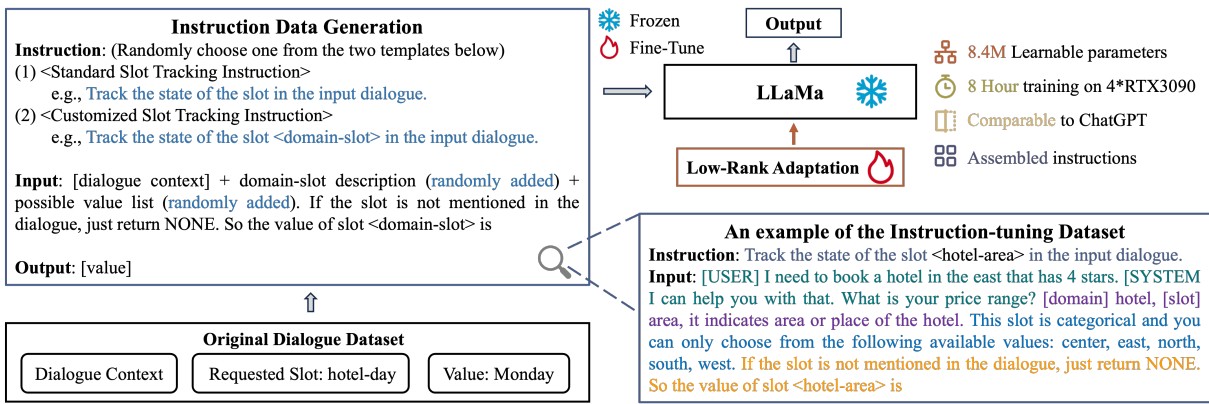

Figure 4: Structure of the LDST model. In the first step, we construct the instruction dataset from the original dataset using the Instruction Data Generation module. Next, we utilize the parameter-efficient fine-tuning technique to train the foundation model with the instruction dataset.

such as the "[USER]" segment token used to indicate the start of a system utterance. And both the domain-slot description prompt and the PVL prompt are supplementary descriptions of the requested slot, they all derive from the given schema in original dataset (PVL prompts are only available for categorical slots). Here, to simulate the situation when the model may not get a description of the requested slot or it's possible values during the testing phase, we add these two prompt templates randomly with a 50% probability, respectively.

**Ouput Prompt** Finally, the output field consists of the corresponding value $V_J^t$ of the requested slot $S_J$. By following the aforementioned process, we obtained a newly and diverse instruction dataset for the next step of fine-tuning the model.

### 3.2 Parameter Efficient Tuning

In this part, we describe how to fine-tune the foundation model using a parameter efficient approach. LDST takes the instruction and input field from the dataset as inputs and retrieves the corresponding slot value $V_J^t$ as output:

$$V_J^t = Decoder(\hat{\mathcal{X}}) \qquad (1)$$

where Decoder indicates that the foundation model (e.g., LLaMa) uses the Transformer-decoder framework, and $\hat{\mathcal{X}}$ denotes the instruction data, i.e., the combination of instruction and input fields.

As shown in Figure 4, to enhance the efficiency of the fine-tuning process and reduce memory requirements, we utilize Low-Rank Adaptation (LoRA) (Hu et al., 2021). LoRA freezes the pretrained model weights and injects trainable rank decomposition matrices into each layer of the Trans-

former architecture, greatly reducing the number of trainable parameters for downstream tasks. For example, in our experiment with LLaMa 7B, the number of learnable parameters is 8.4M, which is only 0.12% of the total model parameters. Denote by the trainable parameters as a weight matrix $W_0 \in \mathbb{R}^{d \times k}$. Unlike traditional methods that directly modify the values of $W_0$, LoRA introduces an additional set of trainable parameters, denoted as $\Delta W$, which are directly injected into the original $W_0$. We represent the update as $W = W_0 + \Delta W = W_0 + BA$, where $B \in \mathbb{R}^{d \times r}$, $A \in \mathbb{R}^{r \times k}$. The rank $r \ll min(d, k)$. During training, $W_0$ is frozen and does not receive any gradient updates, we only update the parameters in $A$ and $B$. Note both $W_0$ and $\Delta W = BA$ are multiplied with the same input, and their respective output vectors are summed coordinate-wise. For the original output $h = W_0 x$, LoRA modified forward pass yields:

$$h = W_0 x + \Delta W x = W_0 x + BA x. \qquad (2)$$

Finally, the learning objective of the generation process in LDST is to minimize the negative log-likelihood of $V_J^t$ given the context $\mathcal{X}_t$ and slot $S_J$:

$$L = -\sum_t^T \sum_j^J \log p\left(V_j^t \mid \mathcal{X}_t, \mathcal{S}_j\right). \qquad (3)$$

## 4 Experiments

### 4.1 Datasets

We conducted experiments using the benchmark datasets for multi-domain task-oriented dialogue, and Table 1 gives detailed statics on these datasets.

| Characteristics | SGD | MultiWOZ 2.2 | MultiWOZ 2.4 |
|---|---|---|---|
| No. of domains | 16 | 8 | 7 |
| No. of dialogues | 16,142 | 8,438 | 8,438 |
| Total no. of turns | 329,964 | 113,556 | 113,556 |
| Avg. turns per dialogue | 20.44 | 13.46 | 13.46 |
| Avg. tokens per turn | 9.75 | 13.13 | 13.38 |
| No. of slots | 215 | 61 | 37 |
| Have schema description | Yes | Yes | Yes |
| Unseen domains in test set | Yes | No | No |

Table 1: Statistics of the datasets used for training in our experiments.

**Schema-Guided Dialogue (SGD)**  SGD (Rastogi et al., 2020) is the most challenging dataset, consisting of over 16,000 conversations between a human-user and a virtual assistant. It encompasses 26 services across 16 domains, such as events, restaurants, and media. Notably, SGD introduces unseen domains in its test set, challenging the generalization ability of the model.

**MultiWOZ 2.2 and MultiWOZ 2.4**  MultiWOZ 2.4 (Ye et al., 2022a) is an updated version on top of MultiWOZ 2.2 (Zang et al., 2020) to improve DST evaluation, and the validation set and test set of MultiWOZ 2.4 have been carefully reannotated. We therefore treat it as a clean dataset for testing. We also conduct experiments on MultiWOZ 2.2 which is known to contain annotation noise. We used this noisy dataset to test the robustness of the model and to analyse the ability of the model to detect annotation errors present in the test set.

### 4.2 Evaluation Metrics

We adopt the following evaluation metrics, consistent with previous works (Ye et al., 2022b): Joint Goal Accuracy (**JGA**) and Average Goal Accuracy (**AGA**). JGA serves as the primary metric for DST evaluation and represents the ratio of dialogue turns for which the entire state is correctly predicted. AGA is the average accuracy of the active slots in each turn. A slot becomes active if its value is mentioned in the current turn and is not inherited from previous turns.

### 4.3 Main Results

We conducted full-scale evaluations of the LLM-driven LDST model in three distinct experimental settings, where the model showed a tremendous performance improvement in both zero-shot and few-shot settings. These findings can provide valuable insights and contribute to the research community through substantial advances in the field of DST. The detailed results are as follows:

| Domain | SGD-baseline | TransferQA | SDM-DST | D3ST | **LDST** |
|---|---|---|---|---|---|
| Messaging | 10.2/20.0 | 13.3/37.9 | 36.6/61.4 | - | **89.6/90.4** |
| Payment | 11.5/34.8 | 24.7/60.7 | 16.5/62.0 | - | **92.3/96.4** |
| Trains | 13.6/63.5 | 17.4/64.9 | 46.7/86.9 | - | **81.0/94.0** |
| Alarm | 57.7/1.8 | 58.3/81.7 | 58.3/87.5 | - | **94.4/96.9** |
| Average | 20.5/34.2 | 25.9/61.8 | 40.4/76.8 | 83.3/- | **89.3/94.4** |

Table 2: Zero-shot results (JGA(%)/AVG(%)) on SGD.

| Domain | TRADE | SUMBT | SimpleTOD | T5DST | D3ST | **LDST** |
|---|---|---|---|---|---|---|
| Attraction | 19.87 | 22.60 | 28.01 | 33.09 | 57.10 | **75.61** |
| Hotel | 13.70 | 19.80 | 17.69 | 21.21 | 22.30 | **63.32** |
| Restaurant | 11.52 | 16.50 | 15.57 | 21.82 | 38.90 | **73.72** |
| Taxi | 60.58 | 59.50 | 59.22 | 65.09 | 79.00 | **91.47** |
| Train | 22.37 | 22.50 | 27.75 | 35.42 | 39.60 | **71.03** |
| Average | 25.76 | 28.18 | 29.65 | 35.20 | 47.38 | **75.03** |

Table 3: Zero-shot results (JGA(%)/AVG(%)) on Multi-WOZ 2.0.

**Zero-shot Cross-domain Results**  Following previous zero-shot settings (Wang et al., 2022c), all models are first trained on some domains and then evaluated on the test-set of the unseen domain. Here we compare with the baseline models that can predict dialogue state on unseen domains: SGD-baseline (Rastogi et al., 2020), TransferQA (Lin et al., 2021a), SDM-DST (Wang et al., 2022a), SUMBT (Lee et al., 2019), SimpleTOD (Hosseini-Asl et al., 2020), T5DST (Lin et al., 2021b) and D3ST method (Zhao et al., 2022).

Tables 2 and 3 highlight the exceptional performance of our approach in zero-shot cross-domain DST. Specifically, on the SGD dataset, LDST achieves a remarkable 6.0% absolute increase in the JGA score when compared to the larger T5-XXL (11B)-based D3ST model, elevating it from 83.3% to an impressive 89.3%. Additionally, the AGA score experiences a substantial surge of 17.6%, escalating from 76.8% to a remarkable 94.4%.

On the MultiWOZ 2.0 dataset, we observe a significant advancement in the average JGA score, surging from 47.38% to 75.03%, reflecting an absolute improvement of 27.65%. Notably, the Payment domain in the SGD dataset displays the most prominent improvement, with the JGA metric soaring from 24.7% to an astounding 92.3%. This remarkable enhancement is attributed to the Payment domain's independence from the source domains. This significant boost not only demonstrates the powerful transfer learning capabilities of the LDST model but also emphasizes its valuable implications for the DST research community.

**Few-shot Results**  In the few-shot settings, we follow the multi-domain scenario from Wu et al.

| Models | MultiWOZ 2.4 | | |
|---|---|---|---|
| | 1% | 5% | 10% |
| DS2-BART | 30.55 | 42.53 | 41.73 |
| DS2-T5 | 36.76 | 49.89 | 51.05 |
| IC-DST GPT-Neo | 17.36 | 29.62 | 34.38 |
| SM2-11b | 40.03 | 51.14 | 51.97 |
| **LDST** | **46.77** | **56.48** | **62.45** |

Table 4: Results (in JGA(%)) of few-shot experiments on MultiWOZ 2.4.

(2019), where randomly select 1%, 5%, and 10% of training dialogues to train, and evaluation is conducted on the full test set of each domain. The evaluation results on MultiWOZ 2.4 are shown in Table 4, where we compare with SOTA few-shot DST methods: DS2-BART (Shin et al., 2022), DS2-T5 (Shin et al., 2022), IC-DST GPT-Neo (Hu et al., 2022), and SM2-11b (Chen et al., 2023).

The results indicate a clear trend: as the amount of training data increases, the performance of all models consistently improves. Notably, our LDST model stands out in this setting. At the 10% data setting, it achieved significant performance gains by elevating the JGA metric from 51.97% to 62.45%, marking an impressive 10.48% absolute improvement. Even at the 5% data setting, our approach surpassed traditional methods that were using 10% of the data. This highlights LDST's remarkable capacity to excel in learning and capturing the core aspects of the task with a smaller dataset.

**Results of Fine-tuning with Full Training Data**
We also evaluate the performance of LDST using the complete training data, and compare it with the following strong baselines, including SGD-baseline (Rastogi et al., 2020), TRADE (Wu et al., 2019), DS-DST (Zhang et al., 2019), TripPy (Heck et al., 2020), Seq2Seq-DU (Feng et al., 2020), MetaASSIST (Ye et al., 2022b), SDP-DST (Lee et al., 2021), TOATOD (Bang et al., 2023b), DiCoS-DST (Guo et al., 2022b), D3ST (Zhao et al., 2022), paDST (Ma et al., 2019). And the results are shown on Table 5.

We initially note significant advancements in recent LLMs like ChatGPT and LLaMa. Notably, our model achieves competitive performance with ChatGPT and even surpasses it on the SGD dataset, particularly excelling in the AGA metric with a score exceeding 98%.

The paDST method has currently achieved

| Methods | Based-model (# Parameters) | MultiWOZ 2.2 | | MultiWOZ 2.4 | | SGD | |
|---|---|---|---|---|---|---|---|
| | | JGA | AGA | JGA | AGA | JGA | AGA |
| SGD-baseline | - | 42.00 | - | - | - | 25.40 | 90.60 |
| TRADE | - | 45.40 | - | 55.05 | - | - | - |
| DS-DST | BERT$_{base}$(110M) | 51.70 | - | - | - | - | - |
| TripPy | BERT$_{base}$(110M) | 53.50 | - | 64.75 | - | - | - |
| Seq2Seq-DU | BERT$_{base}$(110M) | 54.40 | 90.90 | 67.10 | - | 30.10 | 91.00 |
| MetaASSIST | BERT$_{base}$(110M) | - | - | 78.57 | **99.08** | - | - |
| DiCoS-DST | T5$_{base}$(220M) | 61.13 | 98.06 | - | - | - | - |
| TOATOD | T5$_{base}$(220M) | **63.79** | - | - | - | - | - |
| SDP-DST | T5$_{large}$(770M) | 57.60 | **98.49** | - | - | - | - |
| paDST | XLNet$_{large}$(340M) | - | - | - | - | **86.50** | - |
| D3ST | T5$_{XXL}$(11B) | 58.60 | - | 75.90 | - | **86.40** | - |
| ChatGPT | GPT-3.5-Turbo* | 61.52 | 97.86 | **83.16** | 99.27 | 84.81 | **98.46** |
| LLaMa | LLaMa (7B) | 55.37 | 95.71 | 75.13 | 97.58 | 75.32 | 95.83 |
| **LDST (ours)** | LLaMa (7B) | 60.65 | **98.83** | 79.94 | 98.90 | 84.47 | **99.38** |

Table 5: Results of fine-tuning with full training data. - represents the results are not reported in the original paper. * means that the exact number of parameters is uncertain but definitely exceeds 100 billion.

| Transfer | D3ST (T5$_{XXL}$-11B) | LDST (LLaMa-7B) |
|---|---|---|
| SGD → MultiWOZ 2.4 | 28.9 | **31.6** |
| MultiWOZ 2.4 → SGD | 23.1 | **25.9** |

Table 6: Results (in JGA(%)) of cross-dataset transfer between SGD and MultiWOZ 2.4.

SOTA performance on the SGD dataset (with a JGA score of 86.5%), surpassing LDST's 84.47%. However, it's important to note that paDST relies on additional techniques, which contain back-translation between English and Chinese for data augmentation and special manual rules for model predictions. In contrast, LDST relies solely on the default SGD dataset without additional aids. Another SOTA method is D3ST, which uses T5-XXL as backbone model with 11B parameters (our LDST utilizes a 7B model, for outcomes based on different foundational models and different model sizes, please consult Appendix B). D3ST surpasses LDST on the SGD dataset. However, it's noteworthy that LDST outperforms D3ST on Multiwoz 2.2 and 2.4. Additionally, our model demonstrates improved effectiveness when compared to the LLaMa backbone model, underscoring the ongoing importance of fine-tuning LLMs in current research.

**Results of Cross-dataset Transfer** We further performed experiments to assess cross-dataset transfer capabilities, akin to the experiment outlined in Table 4c by Zhao et al. (2022). The JGA results are presented in Table 6, highlighting LDST's exceptional cross-dataset transfer abilities. When compared to the D3ST method, LDST showcases an average improvement of 2.7% in terms of JGA.

| Models | JGA | AGA |
|--------|-----|-----|
| LLaMa (backbone model) | 68.61 | 96.37 |
| + Traditional Instruction Tunnig | 72.87 | 97.98 |
| **+ Ours** | **75.02** | **99.04** |

Table 7: Ablation study. The mean JGA(%) and AGA(%) scores on Multiwoz 2.2, Multiwoz 2.4 and SGD are reported.

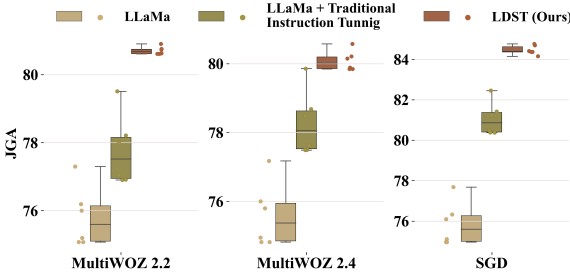

Figure 5: Comparison of the sensitivity of the models to different prompts during the testing phase.

## 4.4 Ablation Study

To validate the effectiveness of the assembled domain-slot instruction tuning, we conducted a comparison with traditional instruction tuning, which employs a fixed prompt template containing all the descriptions for the requested slot (see details in Appendix A.2). The results, as displayed in Table 7, clearly demonstrate that our method outperforms traditional instruction tuning. We observed a substantial 2.15% improvement in the JGA score and a 1.06% improvement in the AGA score.

Next, to analyse the sensitivity of the model to different prompts during the testing phase. we designed six different prompts and evaluated their effects, the results are shown in Figure 5. The results clearly indicate that LDST demonstrates significantly higher accuracy and lower variance in test results compared to the other two baseline methods. The mean variance of our method is 0.04, in contrast to 0.78 for the LLaMa model, representing a substantial decrease of 0.74. These findings highlight that the utilization of the assembled technique for instruction tuning effectively reduces the model's sensitivity to prompts. This not only provides a more stable and efficient inference process but also enhances overall robustness.

## 4.5 Error Analysis

We analyze the types of incorrect predictions in LDST by using the 2835 incorrectly predicted samples on MultiWOZ 2.4. Firstly, 45.72% of the

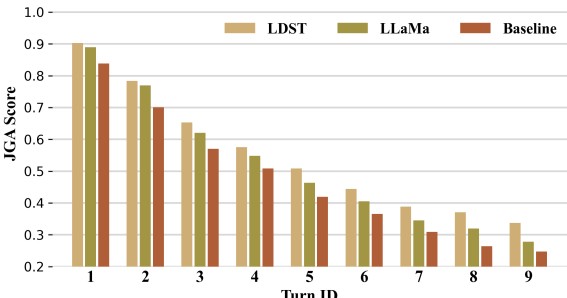

Figure 6: JGA score at each turn on MultiWOZ 2.2.

errors are related to the values "dontcare" and "not mentioned". For example, in cases where the ground truth is "dontcare", the model predicts "not mentioned", and vice versa. Among all 37 slots, the top five with highest error rates are "hotel-type" (338 errors), "restaurant-name" (290 errors), "hotel area" (225 errors), "hotel name" (221 errors), and "attraction name" (205 errors), collectively accounting for 45.11% of the total errors. Specifically, for the "hotel-type" slot, 78.10% of the errors are attributed to the model frequently confusing "not mentioned" with the value "hotel". For instance, the correct value for "hotel-type" was "hotel", but the model incorrectly predicted as "not mentioned".

## 4.6 Effectiveness of LDST in Addressing the Main Challenges of DST

For the co-reference challenge, we analyze the MultiWOZ 2.3 dataset (Han et al., 2021), which includes 253 test samples annotated with co-reference relationships. Our evaluation reveals that the best baseline method achieves an accuracy rate of 91.1%, whereas LDST model achieves an impressive accuracy rate of 96.4%, showcasing the significant improvement offered by our approach.

Additionally, we visualize the JGA score for each dialogue turn in Figure 6 to demonstrate the effectiveness in addressing error propagation. The result clearly shows that as the number of dialogue turns increases, the performance of all methods experiences a decline. However, our LDST model demonstrates a remarkable resilience to error propagation, showcasing a significantly slower decline compared to LLaMa and the best baseline method. These results emphasize the LDST model's capacity to capture and comprehend complex linguistic patterns and dependencies in multi-round conversations, making it a promising solution to mitigate the challenges associated with the DST task.

# 5 Related Work

## 5.1 Multi-Domain Dialogue State Tracking

Recent studies in multi-domain DST have extensively utilized the pre-trained language models to achieve high-performance results (Ravuru et al., 2022; Yu et al., 2022; Sun et al., 2022; Feng et al., 2021; Wang et al., 2022b; Xu et al., 2023c). For example, Xie et al. (2022) proposed a multi-stage correctable dialogue state tracking method to mitigate the error propagation phenomenon, while Wang and Xin (2022) proposed a jointly decision making and a dialogue update technique to prevent error accumulation. In addition, Wang et al. (2022a) and Manotumruksa et al. (2022) solve the challenge of co-referencing by learning correlations between slots, for example, by using a combination of slot prompts or hard copy mechanism. However, these approaches still have limitations, such as the lack of robustness in handling complex dialogue contexts and the difficulty in capturing fine-grained semantic relationships between slots and values.

## 5.2 LLMs for Dialogue State Tracking

While large language models such as GPT-3 (Brown et al., 2020) and T5 (Raffel et al., 2020) have gained significant popularity, the efficient utilization of these models remains a significant challenge. Recent advancements in parameter-efficient fine-tuning (PEFT) techniques have effectively alleviated this problem, such as LoRA(Hu et al., 2021) and Prefix Tuning(Liu et al., 2021). For instance, both Lee et al. (2021) and Yang et al. (2023b) proposed a prompt-tuning method that leverages domain-specific prompts and context information to improve the performance of DST task. Meanwhile, Ma et al. (2023) and Chen et al. (2023) introduced the prefix tuning approach, which involves modifying the input prompt by adding specific tokens at the beginning of the dialogue, aiming to enhance the efficiency of model fine-tuning. However, these methods still face challenges, where the effectiveness heavily depends on prompt design.

Recently, Heck et al. (2023) exclusively tested ChatGPT's performance on the Multiwoz 2.1 dataset. In contrast, our evaluation covers the Multiwoz 2.2, 2.4, and SGD datasets, providing a more comprehensive assessment. While both Pan et al. (2023) and Hudeček and Dušek (2023) included results on the Multiwoz 2.2, Multiwoz 2.4, and SGD datasets, their results were relatively lower due to their use of the text-davinci-003 API. In contrast, we utilized the latest gpt-3.5-turbo API, a highly capable GPT-3.5 model optimized for chat at a fraction of the cost. Consequently, we achieved new SOTA performance with ChatGPT, showcasing its significant strengths.

With the emergence of open-source large language models, such as LLaMa (Touvron et al., 2023), it provides a series of higher-quality backbone models with different parameters. Leveraging LLaMa and the technique of instruction tuning has proven to achieve better results (Taori et al., 2023), opening new avenues for our research.

# 6 Conclusion

In this study, we conduct an initial examination of ChatGPT's capabilities in multi-domain DST, showcasing its superiority over previous methods. This comprehensive evaluation provides useful directions for researchers to design and enhance dialogue systems. To solve the limitations of ChatGPT, we present LDST, an LLM-driven DST framework based on smaller, open-source foundation models. By utilizing a novel assembled domain-slot instruction tuning method, LDST achieves performance on par with ChatGPT. Comprehensive evaluations across three distinct experimental settings demonstrate the remarkable performance improvements achieved by LDST compared to previous methods.

# Limitations

This work has two main limitations: (1) Subjectivity in prompt design: Prompt engineering has shown significant potential for the application of LLMs. However, the prompt designs used in our study are subjective and may not necessarily represent the optimal choices. Consequently, the effectiveness of using these prompts for model fine-tuning or testing may not always yield the best results. Exploring more systematic automated techniques for prompt design could enhance the overall performance of the model. (2) Input length constraints: In our study, we set the input length of the model to 512, which was determined based on statistical analysis and already contains more than 90% of the samples. While it is possible to increase the input length, doing so may result in slower training and inference times. Additionally, when the dialogue or description content becomes too long, the challenge of effectively truncating or summarizing the input arises. Further investigation

into handling longer input sequences without compromising model efficiency would be beneficial.

## Acknowledgments

We thank the anonymous reviewers for their valuable feedback and constructive comments, which greatly contributed to improve the quality of this work. This research was partially supported by the grant of HK ITF ITS/359/21FP.

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

# A  Description of Prompt Templates

## A.1  Prompt Templates for ChatGPT Request

Initially, we noticed that the results reported in the studies by Pan et al. (2023); Hudeček and Dušek (2023) were notably lower in comparison to our results. We attribute this observation to two primary factors, as outlined below.

**Mitigating the Generation of Excessively Lengthy Responses**  ChatGPT often generated answers with excessively detailed explanations, deviating from the expected response format. For example, when asked about the "train-leaveat" slot, ChatGPT might respond with extensive information such as "Monday at 05:16 for the first train and Monday at 16:16 for the last train," whereas the correct response should be simply "05:16." To address this issue, we introduced a prompt that includes "No explanation!" as an instruction to ChatGPT not to provide detailed explanations. Experimental results indicated a significant improvement in answer accuracy through this approach.

**API Version Differences**  Another factor is the utilization of different API versions. The prior works all relied on the text-davinci-003 API, while we utilized a more powerful gpt-3.5-turbo API, a highly capable GPT-3.5 model optimized for chat at a fraction of the cost.

Below we provide specific samples for the four different prompts in Figure 2.

**Prompt type 1: "single return" + "no demo"**
{
**"instruction":** Now you need to perform the task of multi-domain dialogue state tracking. You need to return the value of the slot I'm asking about simply based on the content of the dialogue. No explanation!
**"input":** Input dialogue: [USER] I would like to find a salon. [SYSTEM] In which city do you prefer the salon to be located? [USER] In SFO will be great. [domain] Hotels_2, it indicates A popular service for searching and booking houses for short term stay [slot] number_of_adults, it indicates Number of people for the reservation. This slot is categorical and you can only choose from the following available values: 1, 2, 3, 4, 5. If the slot is not mentioned in the dialogue, just return NONE.
  So the value of slot <Hotels_2-number_of_adults> is
}

**Prompt type 2: "multi return" + "no demo"**
{
**"instruction":** Now you need to perform the task of dialogue state tracking. And the slot schema is in this list [restaurant-area, hotel-name, attraction-name, ...(the remaining slots are omitted here)], which is in a domain-slot format. You need to return the slot and its value in dict format if the value is not none, and no explanation!
**"input":** Input dialogue: [USER] I would like to find a salon. [SYSTEM] In which city do you prefer the salon to be located? [USER] In SFO will be great.
  Please return the value of slot list [restaurant-area, hotel-name, attraction-name, ...(the remaining slots are omitted here)].
}

**Prompt type 3: "single return" + "one demo"**
{
**"instruction":** Now you need to perform the task of multi-domain dialogue state tracking. And I will show you an example and you need to return to the state of the slot I asked about.
**"input":** The example is: Input dialogue: [User]: I need train reservations from norwich to cambridge [SYSTEM]: I have 133 trains matching your request. ...

```
    So the value of slot <train-departure>
is
    And your result should be Norwich.
    The following is the dialogue you need
to test:
    Input dialogue: [USER] I would like to
find a salon.  [SYSTEM] In which city
do you prefer the salon to be located?
[USER] In SFO will be great.  [domain]
Hotels_2, it indicates A popular service
for searching and booking houses for
short term stay [slot] number_of_adults,
it indicates Number of people for the
reservation. This slot is categorical and
you can only choose from the following
available values: 1, 2, 3, 4, 5. If the
slot is not mentioned in the dialogue,
just return NONE. So the value of slot
<Hotels_2-number_of_adults> is
}
```

**Prompt type 4: "multi return" + "one demo"**

```
{
"instruction": Now you need to perform
the task of multi-domain dialogue state
tracking. And I will show you an example
and you need to return the answer strictly
in the format of the example.
"input": The example is: Input dialogue:
[User]: I need train reservations from
norwich to cambridge [SYSTEM]: I have 133
trains matching your request. ...
    Output    result:    Train-Departure:
Norwich,    Train-Arrival:    Cambridge,
...(the  remaining  slots  are  omitted
here)
    And you need to test this example:
    Input dialogue: [USER] I would like to
find a salon. [SYSTEM] In which city do
you prefer the salon to be located? [USER]
In SFO will be great.
    Please   return   the   value   of   slot
list   [restaurant-area,     hotel-name,
attraction-name, ...(the remaining slots
are omitted here)].
}
```

For practical reasons related to API request costs, we conducted tests using these four prompt templates exclusively on the MultiWOZ 2.2 dataset. Subsequent evaluations on the MultiWOZ 2.4 and SGD datasets focused on the most effective tem-

plate, i.e., "single return" + "no demo."

## A.2 Prompt Template for "Traditional" Instruction Tuning

Here, we present the template for traditional instruction tuning, where "traditional" implies the application of instruction tuning directly to the DST task with a *fixed* prompt template. It's important to highlight that this fixed prompt template includes all slot descriptions, such as the domain-slot description and the list of potential values. This fixed prompt is utilized during both the fine-tuning and testing phases.

```
{
"instruction": Track the state of the slot
<restaurant-area> in the input dialogue.
"input":  [Dialogue  context  omitted...]
[Domain]  restaurant,  [Slot]  area,  it
indicates  the  area  or  place  of  the
restaurant. This slot is categorical, and
you can only choose from the following
available  values:   north,  east,  south,
west.  If the slot is not mentioned in
the dialogue, just return NONE. So the
value of slot <restaurant-area> is
"output": north
}
```

## B Additional Results

### B.1 Comparison of ChatGPT with Zero-shot Methods

Essentially, the evaluation of ChatGPT inherently belongs to the zero-shot setting. However, since we found that ChatGPT's results have been comparable to traditional fine-tuning methods, we put its results in Table 5 in the paper. Additionally, we introduce ChatGPT's results from zero-shot settings and the results are shown in table 8 and 9 below.

| Domain | SDM-DST | LDST | **ChatGPT** |
|---|---|---|---|
| Messaging | 36.6 | 89.6 | **92.8** |
| Payment | 16.5 | 92.3 | **94.1** |
| Trains | 46.7 | 81.0 | **83.3** |
| Alarm | 58.3 | 94.3 | **95.7** |
| Average | 40.4 | 89.3 | **91.5** |

Table 8: Zero-shot results (in JGA(%)) of ChatGPT on SGD.

The results clearly demonstrate that ChatGPT outperforms the two strong baselines, SDM-DST and T5DST, by a huge margin. This is primarily

| Domain | T5DST | LDST | **ChatGPT** |
|---|---|---|---|
| Attraction | 33.09 | 75.61 | **78.50** |
| Hotel | 21.21 | 63.32 | **66.75** |
| Restaurant | 21.82 | 73.72 | **77.49** |
| Taxi | 65.09 | 91.47 | **92.38** |
| Train | 35.42 | 71.03 | **72.81** |
| Average | 35.20 | 75.03 | **77.58** |

Table 9: Zero-shot results (in JGA(%)) of ChatGPT on MultiWOZ 2.0.

because the evaluation is conducted in a zero-shot environment, where ChatGPT inherently holds an advantage. It's important to note that as an API service, ChatGPT cannot be tuned offline and is exclusively used for testing purposes.

In the zero-shot setting, the performance of traditional methods (e.g., SDM-DST and T5DST) is worse due to the lack of domain-specific training data. ChatGPT, equipped with its extensive model size and rich pre-trained knowledge, dramatically surpasses the performance of traditional methods and sets the upper bound of performance in the zero-shot setting. It's also worth mentioning that ChatGPT's performance approaches the results of traditional methods fine-tuned on the complete training dataset, which is why we include it in Table 5 for comparison

In contrast, our LDST, utilizing a customized instruction tuning method, effectively approaches ChatGPT's performance in the zero-shot setting, with an average performance difference of 2.4% in the JGA score.

## B.2 Results with Different Foundation Models

We further performed evaluations using an alternative foundation model, Llama2-7B (Touvron et al., 2023), as depicted in Table 10 below.

| Methods | Backbone | Multiwoz 2.4 | SGD |
|---|---|---|---|
| ChatGPT | GPT-3.5-Turbo | 83.2 | 84.8 |
| LLaMa-7B | LLaMa (7B) | 75.1 | 75.3 |
| LDST_LLaMa | LLaMa (7B) | 79.9 | 84.5 |
| LDST_LLaMa2 | LLaMa2 (7B) | 81.9 | 85.1 |

Table 10: Results (in JGA(%)) with different backbones.

The results show that LDST_LLaMa2 performed the best on SGD, attaining a JGA of 85.1% and demonstrating a performance akin to that of ChatGPT on MultiWOZ 2.4. It suggests that a stronger backbone can lead to better DST performance.

## B.3 Results with LLaMa of different sizes

In order to investigate how model size influences performance, we have incorporated supplementary experimental findings involving the LLaMa-13B and -30B models on the SGD dataset. These results are presented in Table 11 below.

| Methods | Backbone | # Training Epochs | SGD |
|---|---|---|---|
| ChatGPT | GPT-3.5-Turbo | n/a | 84.8 |
| D3ST | T5 XXL (11B) | not provided | 86.4 |
| LDST | LLaMa (7B) | 2 | 84.5 |
| LDST | LLaMa (13B) | 2 | 86.5 |
| LDST | LLaMa (30B) | 0.5 | 86.9 |

Table 11: Results (in JGA(%)) with backbones of varying sizes.

The results provide a clear indication that an increase in model size corresponds to an improvement in the JGA score. However, in practical applications, a 7B model not only offers a more suitable fit for local deployment but also showcases impressive performance.

## B.4 Inference Time Analysis

The table 12 below provides the results of inference time. It's worth highlighting that we employ 8-bit quantization for the LLMs, which leads to slower inference times compared to standard 32-bit configurations.

| Methods | Inference Speed (Samples/Min) |
|---|---|
| T5_large-770M | 531 |
| LDST_LLaMa-7B | 90 |
| LDST_LLaMa2-7B | 84 |
| LDST_LLaMa-13B | 64 |
| LDST_LLaMa-30B | 35 |

Table 12: Inference time for different models.

T5 large is the backbone model of the SDP-DST baseline method. From the table, it's clear that the inference speed decreases as the model size increases. For example, LDST_LLaMa-7B predicts an average of 90 samples per minute. When compared to the baseline method based on T5-large (770M), the speed of LDST is approximately one-sixth that of the baseline.

## B.5 Effect of LoRA Configurations

In our work, we utilized common configurations: lora_r = 8 and lora target modules=[query_proj, key_proj, value_proj, output_proj] in each self-attention module that needs to be updated.

To further clarify the impact of LoRA configurations on the experimental results, we performed additional analysis on the Multiwoz 2.4 dataset using 1% training set (To save training time, we set the epoch to 1). We varied the lora_r parameter with values of 1, 2, 4, 8, and 16. In addition, we experimented with two different lora_target_modules configurations: [q_proj, v_proj] and [q_proj, k_proj, v_proj, o_proj]. This resulted in 10 distinct experimental setups.

| lora_target_modules \lora_r | 1 | 2 | 4 | 8 | 16 |
|---|---|---|---|---|---|
| [q proj, v proj] | 29.75 | 31.38 | 33.11 | 39.07 | 40.40 |
| [q proj, k proj, v proj, o proj] | 31.59 | 40.11 | 36.09 | 40.19 | 42.02 |

Table 13: Effect of LoRA configurations. All results are reported in JGA(%).

In these results, a smaller value of "lora_r" indicates fewer LoRA parameters, while lora target modules determines which modules receive LoRA update matrices. Generally, updating more attention matrices yields better results, and performance improves as "lora_r" increases. However, it's essential to note that higher "lora_r" values might extend the model's training time. Hence, selecting appropriate hyperparameters is crucial.

### B.6 Results on MultiWOZ 2.1 Dataset

For a comprehensive evaluation, refer to Table 14, which presents the results on the MultiWOZ 2.1 dataset, comparing ChatGPT by Heck et al. (2023) with the D3ST method by Zhao et al. (2022).

| Methods | Based-model | MultiWOZ 2.1 |
|---|---|---|
| ChatGPT | GPT-3.5-text-davinci-003 | 56.44 |
| D3ST | T5 XXL (11B) | 57.80 |
| LDST (ours) | LLaMa (7B) | 56.69 |

Table 14: Results (in JGA(%)) on MultiWOZ 2.1.

The results reveal that LDST's performance is marginally below that of D3ST. This could be attributed to potential noise in the test set annotations, mirroring our observations regarding the MultiWOZ 2.2 dataset.

## C Implementation Details

### C.1 Data Preprocessing and Evaluation

**Step 1 - Standard Preprocessing** In line with the approach used by Lee et al. (2021), this initial step involves the extraction of dialogue content and slot-value pairs from the raw data. For instance,

consider the dialogue labeled "PMUL4398.json" in the Multiwoz 2.2 training dataset. It comprises 6 dialogue turns between the system and the user. With Multiwoz 2.2 featuring 49 unique slots, this dialogue yields 294 (6*49) training data samples. Here is a specific example:

```
{
"dialogue": [SYSTEM] What can I help you
with [USER] i need a place to dine in the
center thats expensive [SYSTEM] I have
several options for you; do you prefer
African, Asian, or British food? [USER]
Any sort of food would be fine, as long
as it is a bit expensive. Could I get
the phone number for your recommendation?
[domain] restaurant find places to dine
and whet your appetite [slot] area area or
place of the restaurant [Possible Values]
centre, east, north, south, west
"state": centre
}
```

In this example, the "dialogue" field includes the content of the dialogue $(A1, U1), (A2, U2)$, the tracked slot <restaurant-area>, and it's description. The "state" field is the value of the corresponding slot. For the slots that are not mentioned in the dialogue, the "state" field is assigned to NONE.

**Step 2 - Instruction Data Generation** While the preprocessing in Step 1 provided valuable data, it didn't entirely align with the required format for instruction tuning. As a result, it led to suboptimal experimental performance. To address this, we introduced an additional preprocessing stage known as the "Instruction Data Generation Module," as depicted in Figure 4. This module is designed to construct more suitable prompts.

The aforementioned details the entirety of the preprocessing procedure, after which it can be leveraged for both model training and testing.

**Evaluation** Regarding evaluation, we also utilized the code provided by Lee et al. (2021) to calculate JGA and AGA scores. A prediction was considered correct when it exactly matches the ground truth. During the testing phase, we used a consistent prompt template that included domain-slot descriptions and lists of potential values. Experimental findings showed that this template slightly outperformed others because of its provision of more comprehensive slot information.

## C.2 Experimental Settings

During the training phase, we utilized a batch size of 128 and set the learning rate to 1e-4. The number of epochs varied depending on the specific experiment setup. For zero-shot experiments, we trained for 3 epochs. For few-shot experiments, we conducted experiments with different percentages of labeled data: 1% few-shot experiments were trained for 10 epochs, 5% few-shot experiments for 3 epochs, and 10% few-shot experiments for 2 epochs. For fine-tuning on the full dataset experiments, we used 2 epochs.

The model's cutoff length was set at 512, based on data analysis. This length was determined to be optimal as it covered more than 90% of the data. For samples with input lengths exceeding 512 tokens, we truncated them to fit within the cutoff length. Additionally, the parameter "train_on_inputs" was set to false, indicating that the model solely computed the loss based on the final output.

Regarding the hyperparameters of the LORA module, we set the lora rank to 8, alpha to 16, dropout rate to 0.05, and selected "q_proj", "k_proj", "v_proj" and "o_proj" as the LORA target modules. Furthermore, in order to reduce the memory usage of the model, we employed 8-bit quantization techniques to further optimize the fine-tuning process.

We would also like to offer further insights into the training time comparison of our model. In experiments involving fine-tuning on the full dataset, our model had an average training time of 8 hours. In contrast, powerful baseline methods, such as SDP-DST (Feng et al., 2023) and DiCoS-DST (Xu et al., 2023b), required approximately 60 hours for fine-tuning the T5 model based on our testing. This substantial difference in training time underscores the efficiency of our approach. And for the TOATOD (Bang et al., 2023b) method, which also utilizes the PEFT technique, the fine-tuning process only focuses on soft prompts, reducing the overall runtime to 12 hours. This runtime is comparable to our method, demonstrating the effectiveness of our approach compared to traditional methods.

In the case of few-shot experiments, the training time for 1% labeled data was 5 hours, 5% labeled data required 8 hours, and 10% labeled data took approximately 10 hours to train. In contrast, the runtime for zero-shot experiments averaged around 12 hours. It's worth noting that our approach did not exhibit significant runtime improvements compared to traditional methods in these settings. However, it does illustrate that our LLM-driven approach achieves the most substantial performance improvements while still maintaining efficiency. These additional insights into the model's runtime in various experimental setups provide a comprehensive understanding of the time required for training our model and its comparison to other baseline methods.