# OpenReview forum: "Towards LLM-driven Dialogue State Tracking"
_EMNLP/2023/Conference — EMNLP 2023 Main_

### Official Review · Reviewer_GUBv · 2023-07-30

**Soundness:** 4

**Excitement:**

4: Strong: This paper deepens the understanding of some phenomenon or lowers the barriers to an existing research direction.

**Missing References:**

1. Mentioned in Reasons to Reject (1)

2. On L87-L99 cite and attribute these criticisms of ChatGPT, I think sources for these shortcomings of ChatGPT should be cited, in order to provide context for readers.

3. This is contemporary work and doesn’t need to be cited by the authors, but I want to note that LDST’s approach (parameter efficient tuning of LLM) was also explored by Mo et al. 2023 (“Parameter-Efficient Fine-Tuning Method for Task-Oriented Dialogue Systems”, https://doi.org/10.3390/math11143048, published July 2023)

**Paper Topic And Main Contributions:**

This paper presents a study on prompting ChatGPT for TOD benchmarks SGD and MultiWOZ, and demonstrate strong performance. However, they note that ChatGPT is not a reliable backbone to build a TOD system around (“closed source nature, request restrictions, raising data privacy concerns, and lacking local deployment capabilities“), and thus also propose LDST, a framework built around an LLM to perform DST through LoRA finetuning of a smaller open-source pretrained LLM (LLaMa). The authors study the performance of LDST on zero-shot, few-shot, and full-data regimes, and demonstrate SOTA results.

**Questions For The Authors:**

1. On what date was ChatGPT used here? ChatGPT is provided as an API service, and can go through changes over time. This should be included in the paper for reproducibility, or at least versioning.

2. For completeness and ease of citation by future work, would it be possible to also report results on MultiWOZ 2.1? 2.1 is the canonical split for the dataset, and the training data between 2.1 and 2.4 should be the same.

3. I’m a little confused by the results on MultiWOZ 2.2 cross-domain transfer (Table 3), which has higher JGA than the full-data results (Table 5): 60.65 vs. 75.03 JGA. Moreover, the JGA reported on each domain in zero-shot is higher than the average JGA in the full-data setting. Is this correct? Does this mean that additional training data is hurting the performance of LDST? If so, please include additional analysis.

**Reasons To Accept:**

LDST is a novel approach to TOD. While the actual method (LoRa tuning of LLaMa) is not novel and is currently being done for many other NLP tasks, to the best of my knowledge this is the first work to try it on the TOD setting. Moreover, it reports SOTA results on both SGD and MultiWOZ, the canonical TOD benchmarks.

**Reasons To Reject:**

While the results demonstrated by the paper are very strong, I have some issues with the paper:

1. This paper does not cite the correct previous SOTA results. There are much stronger results on SGD than the previous SOTA this paper cites (Feng et al., 2022 on L188): Ma et al. 2020 (https://arxiv.org/abs/1912.09297) and Zhao et al. 2022 (https://arxiv.org/abs/2201.08904) which have 86.5 and 86.4 JGA on SGD respectively. While this is still a few points lower than LDST, it is not as large of an improvement. Both papers also report strong results on MultiWOZ, including the cross-domain transfer on MultiWOZ 2.2 (Table 3), and the few-shot results on MultiWOZ (Table 4) — these should also be included in the comparison.

2. There is not enough detail on how ChatGPT is prompted when described in Section 2. From what I can tell, the authors do not provide any detail on how the prompts are designed other than Figure 2, and there is not enough information to infer more details (e.g. what is “multiple return”? What is “no/one demo”?). This is detrimental to the reproducibility of the prompting results.

    a. Moreover, prior work on prompting ChatGPT for MultiWOZ has had much weaker results (https://arxiv.org/abs/2304.06556, https://arxiv.org/abs/2306.01386, https://arxiv.org/abs/2302.04023). Please cite and compare against these additional results, and provide an analysis on how the results done in this paper differ.

3. While the results demonstrated by LDST are strong, the specific approach it follows (LoRa tuning on LLaMa) is not very original or innovative, and not directly relevant to the TOD setting. Many other groups are working on following this exact approach for many other NLP tasks.

4. It’s also not clear how much of the win that LDST demonstrates is from using a stronger pretrained LLM (LLaMa) over prior works (e.g. Zhao et al. 2022, https://arxiv.org/abs/2201.08904, which uses T5 XXL 11b). This paper would benefit from an additional ablation experiment in which the LLaMa LM is replaced with a different LM, and compared to prior results built off that LM (e.g. GPT-2,3 or T5).

**Reproducibility:**

4: Could mostly reproduce the results, but there may be some variation because of sample variance or minor variations in their interpretation of the protocol or method.

**Reviewer Confidence:**

4: Quite sure. I tried to check the important points carefully. It's unlikely, though conceivable, that I missed something that should affect my ratings.

---

> ### Author Rebuttal · Authors · 2023-08-29
>
> ### Q1. Evaluation date and API version of ChatGPT
>
> >Thank you for your valuable suggestion.
>
> >For the MultiWOZ 2.2 dataset, evaluations were conducted between April 15th and 18th, 2023. The evaluations of MultiWOZ 2.4 occurred between June 10th and 12th, 2023. The SGD dataset was assessed between June 14th and 17th, 2023. All evaluations were performed using the gpt-3.5-turbo API service.
>
> >We will include the above information in the final version to ensure reproducibility.
>
>
> ### Q2. For completeness and ease of citation by future work, would it be possible to also report results on MultiWOZ 2.1?
>
> >Thanks for your suggestion. The table below provides the results on the MultiWOZ 2.1 dataset, where we compared ChatGPT by [Heck et al. (2023)](https://arxiv.org/abs/2306.01386) and the D3ST method by [Zhao et al. (2022)](https://arxiv.org/abs/2201.08904).
>
> | Methods    | Based-model(# Parameters) | MultiWOZ 2.1 |
> |------------|---------------------------|-------------|
> | ChatGPT    | GPT-3.5-text-davinci-003  | 56.44       |
> | D3ST       | T5 XXL (11B)              | 57.80       |
> | LDST (ours)       | LLaMa (7B)                | 56.69       |
>
> >It can be observed that the performance of our LDST is slightly lower than that of D3ST. This may be attributed to noise in the annotation of the test set, a phenomenon similar to our observations on the MultiWOZ 2.2 dataset (lines 193-198). We'll include the results into the paper for completeness and reference by future researchers.
>
>
> ### Q3. Response to confusion about the results on MultiWOZ 2.2 cross-domain transfer (Table 3), which has higher JGA than the full-data results (Table 5)
> >Thanks for your question. This phenomenon is completely normal and occurs due to the different evaluation settings between the zero-shot and full-data experiments.
>
> >In the zero-shot setting (lines 384-387), we focus on evaluating the accuracy of the model within a single domain. On the other hand, in the full-data setting, we calculate the multi-domain JGA score, which requires the model to accurately predict slots across all domains simultaneously."
>
>
> >Take the following ground truth state list as an example, which includes three slots from three different domains. In the zero-shot setting, for instance, when predicting the attraction domain, the model only needs to predict one slot \<attraction-area\>. However, in the full-data fine-tuning setting, the model has to predict three slots, making its task more challenging, thus resulting in lower JGA scores.
> >* Groundtruth state list with 3 domains:\
> (attraction-area = east, hotel-name = wartworth, restaurant-bookday = sunday)
>
> >Additionally, we provide the JGA score **for each domain** in the MultiWOZ 2.2 dataset in the full-data setting:
> >* Attraction: 87.85;  Hotel: 82.40;  Restaurant: 88.75;  Taxi: 96.49;  Train: 88.52;  Average: 88.81.
>
> >All the single-domain JGA scores are much higher than the multi-domain result (60.65) in Table 5, and when compared to the zero-shot results in Table 3, they show an average improvement of 13.78\%.
>
>
> ### Q4. Response to missing references to previous SOTA results on the SGD dataset
> >Thank you for pointing out the missing references. We will include them in our paper to ensure a comprehensive comparison. Here is a comparison with the missing references:
>
> >Firstly, the [paDST method (Ma et al., 2020)](https://arxiv.org/abs/1912.09297) has currently achieved sota performance on the SGD dataset (with a JGA of 86.5), surpassing LDST's 84.47. However, it's important to note that paDST relies on additional techniques, which contain back-translation between English and Chinese for data augmentation and special manual rules for model predictions. In contrast, LDST relies solely on the default SGD dataset without additional aids.
>
> >Another SOTA method is D3ST [(Zhao et al., (2022)](https://arxiv.org/abs/2201.08904), which uses T5-XXL as backbone model with 11B parameters (our LDST utilizes a 7B model). D3ST surpasses LDST on the SGD dataset. However, it's noteworthy that LDST outperforms D3ST on Multiwoz 2.2 and 2.4.
>
> >In addition to adding D3ST (T5-XXL)'s zero-shot experimental results in Tables 2 and 3, we also conducted experiments to evaluate the cross-dataset transfer capabilities, similar to the experiment in Table 4c in the work of [Zhao et al. (2022)](https://arxiv.org/abs/2201.08904). The results in JGA are provided as follows:
> >
> | Transfer      | D3ST (T5-XXL 11B) | LDST (LLaMa-7B) |
> |---------------|-------------------|-----------------|
> | SGD&rarr;MultiWOZ 2.4 | 28.9              | 31.6            |
> | MultiWOZ 2.4&rarr;SGD | 23.1              | 25.9            |
>
> >These results demonstrate LDST's superior cross-dataset transfer capabilities. In comparison to the D3ST method, LDST exhibits an average improvement rate of 2.7\% in terms of JGA score.
>
>
>
> ### Q5. Citing criticisms of ChatGPT.
> >Thank you for your suggestion. We will reference these studies to substantiate the shortcomings of ChatGPT:
> >
> * [Zhou C et al. A comprehensive survey on pretrained foundation models: A history from bert to chatgpt](https://https://arxiv.org/abs/2302.09419)
> * [Yang X et al. Exploring the limits of chatgpt for query or aspect-based text summarization](https://https://arxiv.org/abs/2302.08081)
> * [Cao Y et al. A comprehensive survey of ai-generated content (aigc): A history of generative ai from gan to chatgpt](https://https://arxiv.org/abs/2303.04226)
>
>
> ### Q6. Response to the evaluation details of ChatGPT
> >We appreciate your feedback and are committed to improving the clarity of our paper's expressions. In response, we have made the following modifications to Section 2 to enhance clarity:
> >* **Clarification on Single Return and Multi Return**\
> In Figure 2, "single return" and "multi return" refer to the number of slot values returned in each ChatGPT API request. "Single return" involves requesting and receiving values for one slot at a time, while "multi return" entails requesting and receiving values for all slots simultaneously. For instance, in the MultiWOZ 2.2 dataset which has 49 different slots, "multi return" retrieves values for all 49 slots in a single request.
> This causes a significant increase API requests for "single return" but simplifies the model's task, resulting in improved performance. Conversely, "multi return" reduces API requests but increases token count per request.
> >* **Explanation of No/One Demo**\
> "No/one demo" denotes whether an example is provided in the prompt as a demonstration to aid the model's comprehension of the task. Selecting "one demo" is similar to adopting the in-context learning concept.
>
> >We will also include details on all four prompt template variations in the Appendix section.
>
> >Below is an example illustrating the case of "multi return" + "one demo":
> >* Prompt type: "multi return" + "one demo"\
> \{ "**instruction**": "Now you need to perform the task of multi-domain dialogue state tracking. The slot schema is in this list [restaurant-area, hotel-name, attraction-name, ...(the remaining slots are omitted here)], which is in a domain-slot format. I will show you an example and you need to return the answer strictly in the format of the example.", \
> "**input**": "The example is: Input dialogue: [User]: I need train reservations from norwich to cambridge [SYSTEM]: I have 133 trains matching your request.  ...
> Output result: Train-Departure: Norwich, Train-Arrival: Cambridge, Train-Departure-Time: Monday, Attraction-Name: cineworld cinema ...
> And you need to test the following example:
> Input dialogue:  [USER] I would like to find a salon. [SYSTEM] In which city do you prefer the salon to be located? [USER] In SFO will be great. ...
> Please return the value of slot list [restaurant-area, hotel-type, attraction-name, ...(the remaining slots are omitted here)], where restaurant-area means area or place of the restaurant, hotel-type means what is the type of the hotel, ...  " \}
>
> >For practical reasons related to API request costs, we conducted tests using these four prompt templates exclusively on the MultiWOZ 2.2 dataset. Subsequent evaluations on the MultiWOZ 2.4 and SGD datasets focused on the most effective template, i.e., "single return" + "no demo."
>
>
> ### Q7. Response to comparison with evaluation results of ChatGPT in other papers
> >We appreciate your feedback and will integrate the discussion into the related work section. Below is a comparison of our results with those from other studies:
>
> >[Heck et al. (2023)](https://arxiv.org/abs/2306.01386) exclusively tested ChatGPT's performance on the MultiWOZ 2.1 dataset. In contrast, our evaluation covers the MultiWOZ 2.2, MultiWOZ 2.4, and SGD datasets, providing a more comprehensive assessment. While [Pan et al. (2023)](https://arxiv.org/abs/2304.04256), [Bang et al. (2023)](https://arxiv.org/abs/2302.04023) and [Hudeček et al. (2023)](https://arxiv.org/abs/2304.06556) presented results for the MultiWOZ 2.2, MultiWOZ 2.4, and SGD datasets, we observed that their reported results were comparatively lower. We attribute this observation to two main factors:
> >* **Mitigating Common ChatGPT Errors**\
> ChatGPT often provided answers with excessive explanations, diverging from expected formats. For instance, when prompted for the "train-leaveat" slot, ChatGPT might respond with overly detailed information, like "Monday at 05:16 for the first train and Monday at 16:16 for the last train", while the correct answer should be simply "05:16."
> To address this issue, we added "No explanation!" in the prompt. This prompt instructs ChatGPT not to explain its answers. Experimental results showed that this significantly enhances answer accuracy.
> >* **API Version Differences**\
> Another factor is the utilization of different API versions. The prior works all relied on the text-davinci-003 API, while we utilized a more powerful gpt-3.5-turbo API, a highly capable GPT-3.5 model optimized for chat at a fraction of the cost.
>
> >Consequently, we achieved new SOTA performance with ChatGPT, showcasing its significant strengths. Moreover, compared to these papers, our work's primary contribution lies in fine-tuning open-source LLMs and evaluating their performance across three distinct experimental settings, providing valuable insights for the DST community.
>
>
> ### 8. Response to additional ablation experiments on the backbone model
> >We appreciate your suggestions and have conducted additional experiments using various backbone models, including [Llama2-7B (Touvron et al., 2023)](https://arxiv.org/abs/2307.09288) and [Llama-13B (Touvron et al., 2023)](https://[arxiv.org/abs/2307.09288](https://arxiv.org/abs/2302.13971)). We also included results for a T5-XXL-based method, D3ST  [(Zhao et al., 2022)](https://arxiv.org/abs/2201.08904), and the results are as follows:
>
> | Methods                              | Based-model(# Parameters) | Multiwoz2.4 | SGD  |
> |--------------------------------------|---------------------------|-------------|------|
> | ChatGPT                              | GPT-3.5-Turbo             | 83.2        | 84.8 |
> | D3ST                                 | T5-XXL (11B)              | 75.9        | 86.4 |
> | LDST (Ours)                           | LLaMa (7B)                | 79.9        | 84.5 |
> |                                      | LLaMa2 (7B)               | 81.9        | 85.1 |
> |                                      | LLaMa (13B)               | 82.4        | 86.2 |
>
> >LDST_LLaMa-13B exhibits performance comparable to ChatGPT on the MultiWOZ 2.4 dataset and  achieves similar performance on SGD compared to D3ST. Moreover, our LDST outperforms D3ST on both Multiwoz 2.2 (see Table 5 in the paper) and 2.4 datasets by using only a 7B backbone model. These results underscore the robustness of LDST across different backbone models.

---

### Official Review · Reviewer_Qbz6 · 2023-08-01

**Soundness:** 4

**Excitement:**

4: Strong: This paper deepens the understanding of some phenomenon or lowers the barriers to an existing research direction.

**Paper Topic And Main Contributions:**

This paper proposes a new method to perform Dialogue State Tracking (DST). DST performance is measured by the Joint Goal Accuracy on the Schema-Guided Dialogues (SGD) and MultiWoz (versions 2.2, 2.3, and 2.4) datasets, as is typically done in the literature. They evaluate their approach in the zero-shot, few-shot, and full data settings.

They propose a method called LDST that consists in training a LoRA adapter for a large language model (LLaMa; Touvron et al., 2023) module on an instruction dataset build off of the original training set of the benchmark datasets. To build the instruction-tuning datasets, they propose the Assembled Domain-Slot Instruction Generation method. This approach generates diverse instruction samples by randomly combining different instruction and input templates, exposing the model to diverse instruction types during the fine-tuning process to reduce the model’s sensitivity to prompts. They compare their method with some strong baselines from the literature and their application of ChatGPT to the DST task.

The main contribution is proposing the Assembled Domain-Slot Instruction Generation method to create an instruction finetuning dataset.

**Questions For The Authors:**

Questions:
- Results on these datasets are very, _very_, dependent on how the data are preprocessed; can you expand on how you preprocessed the labels and performed the evaluation (can you add more details in the appendix)? (this is the most important of my questions and I am happy to increase the soundness score after the authors' response)
- How much of the results depend on the backbone model (i.e., Llama)? It would be great to add evaluation with other backbones
- How does performance scale with the size of the Llama model?
- How much longer does Llama + adapter take to perform inference? (useful for practitioners)
- What's the effect of the LoRA configurations?

Suggestions:
- Many of the baselines feature much smaller models (<1B) vs Llama. You could include the T5-XXL (11B) results from [D3ST (Zhao et al., 2022)](https://arxiv.org/abs/2201.08904) which can help you to argue that your approach is competitive with models with similar sizes. Also, it's a good comparison to encoder-decoder models.
- Remove the focus from ChatGPT. As the paper is currently written, there is a lot of emphasis on ChatGPT. Given the good results of your methodology, I think ChatGPT can be reprioritised to be a nice-to-have evaluation.
- Abstract can be shortened
- Highlight the efficiency gains (time and computation) in training vs baselines
- More structured appendix featuring the data preprocessing and evaluation process (possibly with examples)
- I think the appendix includes some interesting info that would benefit the community (e.g., training details, quantisation). Usually, these would not be interesting for typical backbones (e.g, T5), however, for these new LLMs I think this would provide good insights for the community.

**Reasons To Accept:**

- Methodology is simple and effective (good performance wrt baselines).
- Evaluation is thorough (zero-shot, few-shot, and full data settings) and performed on the 2 main benchmarks in this literature (SGD and MultiWoz)
-  Interesting insights on the applications of LLM to DST (more in the Questions section below)

**Reasons To Reject:**

Currently, the data preprocessing and evaluation are not explained in depth. Since results are very much dependent on these factors, the paper needs to add more details about it to improve its soundness (happy to raise the soundness score once addressed).

**Reproducibility:**

4: Could mostly reproduce the results, but there may be some variation because of sample variance or minor variations in their interpretation of the protocol or method.

**Reviewer Confidence:**

4: Quite sure. I tried to check the important points carefully. It's unlikely, though conceivable, that I missed something that should affect my ratings.

**Typos Grammar Style And Presentation Improvements:**

- Some citations are concatenated with the text, e.g. "some things to say(CITATION, 2023)"
- I would suggest removing hype-y adjectives (e.g., "exceptional", etc)
- "mains" -> "main" in subsection title 4.6

---

> ### Author Rebuttal · Authors · 2023-08-29
>
> ### Q1. Results on these datasets are very dependent on how the data are preprocessed; can you expand on how you preprocessed the labels and performed the evaluation (can you add more details in the appendix)?
>
> >Firstly, we greatly appreciate your question and the opportunity to clarify our data preprocessing and evaluation procedures. Please note that we did not make any additional changes to the labels, just followed the well-established procedure outlined by [Lee et al., (2021)](https://arxiv.org/abs/2109.07506). The steps include:
>
>
> >**Data Preprocessing**
> >* **Step 1 - Standard Preprocessing**\
> Same as [Lee et al., (2021)](https://arxiv.org/abs/2109.07506), this step extracts dialogue content and slot-value pairs from the raw data. \
> >For instance, consider the dialogue labeled "PMUL4398.json" in the Multiwoz 2.2 training dataset. It comprises 6 dialogue turns between the system and the user. With Multiwoz 2.2 featuring 49 unique slots, this dialogue yields 294 (6*49) training data samples.\
> >Here is a specific example:\
> \{ "**dialogue**": "[SYSTEM] What can I help you with [USER] i need a place to dine in the center thats expensive [SYSTEM] I have several options for you; do you prefer African, Asian, or British food? [USER] Any sort of food would be fine, as long as it is a bit expensive. Could I get the phone number for your recommendation? [domain] restaurant find places to dine and whet your appetite [slot] area area or place of the restaurant [Possible Values] centre, east, north, south, west", \
> "**state**": "centre" \} \
> In this example, the "dialogue" field includes the content of the dialogue \{(A1,U1), (A2,U2)\}, the tracked slot \<restaurant-area\>, and it's description; the "state" field is the value of the corresponding slot. For the slots that are not mentioned in the dialogue, the "state" field is assigned to NONE.
> >* **Step 2 - Instruction Data Generation**\
> >Although the preprocessing in Step 1 yielded valuable data, this data didn't completely conform to the required format for instruction tuning, resulting in subpar experimental performance. Hence, we introduced an additional preprocessing stage, referred to as the "Instruction Data Generation Module" in Figure 4, to construct more appropriate prompts.
>
> >The aforementioned details the entirety of the preprocessing procedure, after which it can be leveraged for both model training and testing.
>
>
> >**Evaluation**\
> Regarding evaluation, we also utilized the code provided by [Lee et al., (2021)](https://arxiv.org/abs/2109.07506) to calculate JGA and AGA. A prediction is considered correct when it exactly matches the ground truth.
>
> >During the testing phase, we employed a fixed prompt template with domain-slot descriptions and possible value lists. Experimental results indicated that it outperformed other templates slightly, as it provided more comprehensive slot information.
> >
> >Additionally, in Figure 5, we compared the impact of different templates on model performance to better assess the model's sensitivity to prompts. The detailed steps and specific examples will be provided in the appendix for reference.
>
> ### Q2. How much of the results depend on the backbone model (i.e., Llama)? It would be great to add evaluation with other backbones.
> >We appreciate your suggestion. In the paper, we initially showcased the performance of the original LLaMa in Table 5, which approached but did not match SOTA methods. However, the performance can be significantly enhanced with our fine-tuned LDST model.
>
> >Additionally, we conducted evaluations with the recently released open-source LLM [Llama2-7B (Touvron et al., 2023)](https://arxiv.org/abs/2307.09288), illustrated in the table below. (Due to time constraints, we only tested on the SGD and Multiwoz 2.4 datasets. We will include the results on the remaining Multiwoz 2.2 datasets in the final version.)
>
> | Methods                              | Based-model(# Parameters) | Multiwoz 2.4 | SGD  |
> |--------------------------------------|---------------------------|-------------|------|
> | ChatGPT                              | GPT-3.5-Turbo             | 83.2        | 84.8 |
> | LLaMa-7B                             | LLaMa (7B)                | 75.1        | 75.3 |
> | LDST_LLaMa  | LLaMa (7B)    | 79.9        | 84.5 |
> | LDST_LLaMa2                 | LLaMa2 (7B)               | 81.9        | 85.1 |
>
> >The results show that LDST_LLaMa2 performed the best on SGD, attaining a JGA of 85.1% and demonstrating a performance akin to that of ChatGPT on MultiWOZ 2.4. It suggests that a stronger backbone can lead to better DST performance.
>
>
>
> ### Q3. How does performance scale with the size of the Llama model?
> >Thank you for your question. To explore the impact of model size on performance, we have included additional experimental results using the LLaMa-13B and -30B models on SGD dataset. Here are the results:
> >
> | Methods    | Based-model(# Parameters) | SGD  |
> |------------|---------------------------|------|
> | ChatGPT    | GPT-3.5-Turbo             | 84.8 |
> | LDST       | LLaMa (7B)                | 84.5 |
> | LDST       | LLaMa (13B)               | 86.2 |
> | LDST       | LLaMa (30B)               | 86.9 |
>
> >The results in the table clearly show that as the model size increases, the JGA score improves. Nevertheless, in practical terms, a 7B model not only fits better for local deployment but also demonstrates commendable performance.
> >
>
>
> ### Q4. How much longer does Llama + adapter take to perform inference?
>
> >Thanks for your suggestion. The table below provides the results of inference time. It's important to note that we utilize 8-bit quantization for the LLMs, leading to slower inference compared to the standard 32-bit configuration.
>
> | Methods     | T5_large-770M | LDST_LLaMa-7B |  LDST_LLaMa2-7B | LDST_LLaMa-13B | LDST_LLaMa-30B |
> |-------------|-------------------------|----------------|----------------|----------------|----------------|
> | Samples/Min | 531                     | 90            |  84             | 64             | 35             |
>
> >(T5 large is the backbone model of the [SDP-DST](https://arxiv.org/abs/2109.07506) baseline method.)
>
> >From the table, it's clear that the inference speed decreases as the model size increases. For instance, LDST_LLaMa-7B predicts an average of 90 samples per minute. When compared to the baseline method based on T5-large (770M), the speed of LDST is roughly a sixth of that of the baseline.
> >
>
>
>
> ### Q5. What's the effect of the LoRA configurations?
> >Thank you for your suggestion. In our work, we employed a commonly used configuration: lora\_r = 8 and lora\_target\_modules=[query\_projector, key\_projector, value\_projector, output\_projector] in each self-attention module that needs to be updated.
> >
> >To further clarify the impact of LoRA configuration on the experimental results, we performed additional analysis on the Multiwoz 2.4 dataset using 1\% training set (To save training time, we set the epoch to 1). We varied the lora\_r parameter with values of 1, 2, 4, 8, and 16. In addition, we experimented with two different lora\_target\_modules configurations: [q\_proj, v\_proj] and [q\_proj, k\_proj, v\_proj, o\_proj]. This resulted in 10 different experimental setups.
>
> | lora_target_modules \ lora_r     | 1     | 2     | 4     | 8    | 16   |
> |----------------------------------|-------|-------|-------|------|------|
> | [q proj, v proj]                 | 29.75 | 31.38 | 33.11 | 39.07 | 40.40 |
> | [q proj, k proj, v proj, o proj] | 31.59 | 40.11 | 36.09 | 40.19 | 42.02 |
>
> >In these results, a smaller "lora_r" indicates fewer LoRA parameters, while "lora_target_modules" determines which modules receive LoRA update matrices. Generally, updating more attention matrices yields better results, and performance improves as "lora_r" increases. However, increasing "lora_r" might prolong model training time, so choosing the appropriate hyperparameters is crucial.
>
>
> ### Q6. Response to Suggestions
> >Thank you very much for your valuable suggestions. We will incorporate the supplementary content mentioned above into the appendix of the paper, presenting it with the help of figures and tables to enhance its comprehensiveness. Additionally, we will rectify any typos present in the manuscript. Once again, we appreciate your feedback, which has contributed to improving the quality and clarity of our work.

---

### Official Review · Reviewer_1Gie · 2023-08-03

**Typos Grammar Style And Presentation Improvements:** 1. Better add ChatGPT results in Tabl…
**Soundness:** 3

**Excitement:**

4: Strong: This paper deepens the understanding of some phenomenon or lowers the barriers to an existing research direction.

**Missing References:**

There are multiple papers that study the effectiveness of ChatGPT on task-oriented dialog modeling, including DST, e.g.
[1] Michael Heck et al., ChatGPT for Zero-shot Dialogue State Tracking: A Solution or an Opportunity?
[2] Wenbo Pan et al., A Preliminary Evaluation of ChatGPT for Zero-shot Dialogue Understanding
[3] Vojtěch Hudeček, Are LLMs All You Need for Task-Oriented Dialogue?
It would be great to compare with these papers.

**Paper Topic And Main Contributions:**

This paper proposes a way to instruction-finetune Llama model for dialog state tracking (DST) tasks. The paper shows that a properly finetuned Llama 7B model works on-par or better than ChatGPT on several DST benchmarks.

**Questions For The Authors:**

1. The instruction directly asks the LM to "perform the tasks of dialog state tracking". I wonder whether ChatGPT or Llama has a good understanding of what "dialog state tracking" is. Therefore, it would be good to add some explanations to what DST is so that LM can more easily follow the instruction.

2. In the design of input prompts (line 278), it seems only categorical slots are being considered (list of possible values). What about non-categorical slots like date, address?

3. The paper argues that the proposed instruction tuning is better than "traditional" instruction tuning (Table 6). What precisely is the definition of "traditional" instruction tuning? Better give some examples, and share some insights into why different instruction tuning methods lead to different performance.

4. What's the comparison between LoRA tuning full model tuning? Better add a comparison.

**Reasons To Accept:**

The paper demonstrates an instruction-tuning method for LM on DST tasks. It also shows that such a tuned model can outperform ChatGPT on DST tasks.

**Reasons To Reject:**

The proposal (instruction tuning) is a fairly standard and widely adopted approach, and the novelty of the paper is limited.
Comparison with other existing work on using ChatGPT for DST is missing.

**Reproducibility:**

5: Could easily reproduce the results.

**Reviewer Confidence:**

4: Quite sure. I tried to check the important points carefully. It's unlikely, though conceivable, that I missed something that should affect my ratings.

---

> ### Author Rebuttal · Authors · 2023-08-29
>
> ### Q1. About whether LMs truly understand the task of "dialog state tracking" and suggests providing explanations to enhance comprehension.
>
> >We're happy to explain this concern. In preliminary stages of the experiment, both ChatGPT and LLaMa demonstrated a good grasp of DST-related questions and provided correct explanations. This confirms their understanding of the "perform the task of multi-domain dialog state tracking" instruction.
>
> >To refine the prompt, an inverse prompting approach was employed. It involved providing LLMs with DST task inputs and outputs, enabling them to generate suitable request prompts. This process contributed to the chosen instruction prompt.
>
> >Having ensured that the LLMs understand the essence of DST, a key factor affecting the model's performance is the correct understanding of the meaning of slots. This is because slots do not usually reflect standard written English, e.g., " arriveby" and "ref". These custom meaning representations are typically abbreviated and/or under-specified, which creates a barrier to the effective utilization of pre-trained LMs.
>
> >To address this issue, we incorporated natural language descriptions from schema files into DST prompts, e.g., the description of “ref” is expanded to “reference number of the hotel booking”. This step aims to bridge the gap between custom slot representations and language model understanding.
>
>
> ### Q2. In the design of input prompts (line 278), it seems only categorical slots are being considered (list of possible values). What about non-categorical slots like date, address?
>
> >Non-categorical slots cannot provide a possible value list (PVL) because their values are context-dependent and can have an infinite range, making PVL prompts impractical for them.
>
> >Historically, categorical and non-categorical slots were treated differently. Categorical slots were treated as classification tasks with N classes based on PVL. For non-categorical slots, strategies such as the copy mechanism were employed to extract values from dialogues and pinpoint value positions.
>
> >In the era of pre-trained language models, both slot types are now unified as generation tasks. Additionally, to ensure accuracy in categorical slots, an extra PVL is provided.
>
> ### Q3. About the definition of "traditional" instruction tuning and why different instruction tuning methods result in varying performance.
>
> >"Traditional" refers to applying instruction tuning directly to the DST task using a **fixed** prompt template. This template includes the standard slot tracking instruction from Fig. 4 and adds both Domain-Slot Description prompt and Possible Values List prompt to each example. This will be clarified in the final version.
>
> >In contrast, our proposed method, "Assembled Domain-Slot Instruction Tuning," is designed for complex multi-domain DST task. It generates diverse and highly specific instruction samples by randomly assembling different instruction prompts and input prompts.
>
> >To illustrate the effectiveness of our method, we present three examples. These examples vary in the information provided in the input prompts, showcasing the versatility of our approach:
>
> >***Training Example A*** (a categorical slot, with slot description and PVL):\
> [Dialogue context omitted...]
> [Domain] restaurant, [Slot] area, it indicates the area or place of the restaurant. This slot is categorical, and you can only choose from the following available values: north, east, south, west. If the slot is not mentioned in the dialogue, just return NONE. So the value of slot \<restaurant-area\> is
>
> >***Training Example B*** (a categorical slot, without slot description and PVL):\
> [Dialogue context omitted...]
> [Domain] train, [Slot] has\_internet. So the value of slot \<restaurant-has\_internet\> is
>
> >***Testing Example C*** (a slot unseen during training, no slot description available):\
> [Dialogue context omitted...]
> [Domain] hotel, [Slot] ref. So the value of slot \<hotel-ref\> is
>
> >**Explanation for the Superior Performance of Our Method:**
> >* **Enhancing model generalization ability to slots**\
> >Traditional tuning (with a fixed prompt template) relies heavily on slot descriptions where the training samples are all of type A, making the model dependent on this information. Our method introduces diverse training samples, including those without any descriptions (Type B). This forces the model to learn slot meanings by analyzing correct outputs, reducing prediction bias when encountering new, undescribed slots during testing. The experiments in Table 6 and Figure 5 confirm the effectiveness of our approach.
> >* **Reducing model sensitivity to prompts**\
> Different prompts can significantly impact model performance. Our approach exposes the model to various prompt variations during training, improving prediction stability. As illustrated in Figure 5, our LDST method yields more consistent results with smaller variances when tested with various templates.
>
> >We also experimented with different backbone models of various sizes ([Llama2-7B (Touvron et al., 2023)](https://arxiv.org/abs/2307.09288), [Llama-13B (Touvron et al., 2023)](https://[arxiv.org/abs/2307.09288](https://arxiv.org/abs/2302.13971)), and [Vicuna-7B (Chiang et al., 2023)](https://lmsys.org/blog/2023-03-30-vicuna)), confirming our LDST method's effectiveness across various settings. LDST_LLaMa-13B achieved the best result on the SGD dataset with a JGA score of 86.2%. We will incorporate the results into the appendix for reader reference.
> >
>
>
> | Methods                              | Based-model(# Parameters) | Multiwoz2.4 | SGD  |
> |--------------------------------------|---------------------------|-------------|------|
> | ChatGPT                              | GPT-3.5-Turbo             | 83.2        | 84.8 |
> | LDST | LLaMa (7B)                | 79.9        | 84.5 |
> |                                      | Vicuna (7B)               | 77.1        | 85.0 |
> |                                      | LLaMa2 (7B)               | 81.9        | 85.1 |
> |                                      | LLaMa (13B)               | 82.4        | 86.2 |
>
>
>
> ### Q4. What's the comparison between LoRA tuning full model tuning? Better add a comparison.
>
> >Thanks for your suggestion. In our study, we did consider full model fine-tuning. However, due to the immense number of parameters in LLMs, executing full fine-tuning on lab-level GPUs (i.e., NVIDIA RTX 3090 we use) becomes quite challenging. Nevertheless, we still try our best to conduct a comprehensive ablation study on LoRA tuning, especially for rank r and target modules. We report results on MultiWOZ 2.4 with 1% training data, summarized in the table below:
>
>
> | lora_target_modules \ lora_r     | 1     | 2     | 4     | 8    | 16   |
> |----------------------------------|-------|-------|-------|------|------|
> | [q proj, v proj]                 | 29.75 | 31.38 | 33.11 | 39.07 | 40.40 |
> | [q proj, k proj, v proj, o proj] | 31.59 | 40.11 | 36.09 | 40.19 | 42.02 |
>
> >In these results, a smaller "lora_r" indicates fewer LoRA parameters, while "lora_target_modules" determines which modules receive LoRA update matrices. Generally, updating more attention matrices yields better results, and performance improves as "lora_r" increases. However, increasing "lora_r" might prolong model training time, so choosing the appropriate hyperparameters is crucial.
>
>
> ### Q5. About Missing References.
>
> >We appreciate the information about other relevant studies, and we'll incorporate the below discussion into the related work section.
>
> >[Heck et al. (2023)](https://arxiv.org/abs/2306.01386) exclusively tested ChatGPT's performance on the Multiwoz 2.1 dataset. In contrast, our evaluation covers the Multiwoz 2.2, 2.4, and SGD datasets, providing a more comprehensive assessment. While both [Pan et al. (2023)](https://arxiv.org/abs/2304.04256) and [Hudeček et al. (2023)](https://arxiv.org/abs/2304.06556) included results on the Multiwoz 2.2, Multiwoz 2.4, and SGD datasets, their results were relatively lower due to their use of the text-davinci-003 API. In contrast, we utilized the latest gpt-3.5-turbo API, a highly capable GPT-3.5 model optimized for chat at a fraction of the cost. Consequently, we achieved new SOTA performance with ChatGPT, showcasing its significant strengths.
>
> >Moreover, compared to these papers, our work's primary contribution lies in fine-tuning open-source LLMs and evaluating their performance across three distinct experimental settings, providing valuable insights for the DST community.
>
>
> ### Q6. Other questions.
>
> >We will add ChatGPT's results to Tables 2 and 3 as follows:
> >
> Table2: Zero-shot results on SGD dataset (JGA score)
> | Domain    | SDM-DST | ChatGPT | LDST (ours) |
> |-----------|---------|---------|------|
> | Messaging | 36.6    | 92.8    | 89.6 |
> | Payment   | 16.5    | 94.1    | 92.3 |
> | Trains    | 46.7    | 83.3    | 81.0 |
> | Alarm     | 58.3    | 95.7    | 94.3 |
> | Average   | 40.4    | 91.5    | 89.3 |
>
> Table3：Zero-shot results on MultiWOZ 2.2 dataset (JGA score)
> | Domain     | T5DST | ChatGPT | LDST (ours)  |
> |------------|-------|---------|-------|
> | Attraction | 33.09 | 78.50   | 75.61 |
> | Hotel      | 21.21 | 66.75   | 63.32 |
> | Restaurant | 21.82 | 77.49   | 73.72 |
> | Taxi       | 65.09 | 92.38   | 91.47 |
> | Train      | 35.42 | 72.81   | 71.03 |
> | Average    | 35.20 | 77.58   | 75.03 |
>
> >From the results, it's evident that ChatGPT surpasses the two baselines, SDM-DST and T5DST, by a huge margin. This is mainly because the assessment is conducted in a zero-shot setting, where ChatGPT inherently has an advantage. As an API service, ChatGPT cannot be tuned offline and is exclusively used for testing purposes.
>
> >In the zero-shot setting, the performance of traditional methods (e.g., SDM-DST and T5DST) is worse due to the lack of domain-specific training data. ChatGPT, with its vast model size and rich pre-trained knowledge, dramatically surpasses the performance of traditional methods and sets the upper bound of performance in the zero-shot setting (Also note that ChatGPT's performance approaches the results of traditional methods fine-tuned on the full training dataset, hence we include it in Table 5 for comparison).
>
> >In comparison, our LDST, with a customized instruction tuning method, effectively approaches ChatGPT's performance in the zero-shot setting, with an average gap of 2.4% JGA score.
>
> >Additionally, the three datasets in Table 6 refer to Multiwoz 2.2, Multiwoz 2.4 and SGD, and we will update the table title accordingly.

---

### Meta-Review · Area_Chair_CYio · 2023-09-19

**Recommendation:** 5

**Metareview:**

The paper presents an instruction-tuning method for applying LLMs to the DST task. To build the instruction-tuning datasets, it proposes the Assembled Domain-Slot Instruction Generation method. This method generates diverse instruction samples by randomly combining different instruction and input templates, which helps to reduce the sensitivity to prompts during the fine-tuning process. This paper performs evaluations in the zero-shot, few-shot, and full data settings and the results are very strong.

Initially, the paper fell short in not comparing against other DST works using ChatGPT. This issue has been addressed in the rebuttal, together with further results on different backbone models. The authors are encouraged to add these new results and all implementation details in their final version.

---

### Decision · Program_Chairs · 2023-10-07

**Decision:**

Accept-Main

**Comment:**

The paper presents an instruction-tuning method for applying LLMs to the DST task. To build the instruction-tuning datasets, it proposes the Assembled Domain-Slot Instruction Generation method. This method generates diverse instruction samples by randomly combining different instruction and input templates, which helps to reduce the sensitivity to prompts during the fine-tuning process. This paper performs evaluations in the zero-shot, few-shot, and full data settings and the results are very strong.

Initially, the paper fell short in not comparing against other DST works using ChatGPT. This issue has been addressed in the rebuttal, together with further results on different backbone models. The authors are encouraged to add these new results and all implementation details in their final version.